# Transverse Momentum Transfer Distributions Following Single Ionization in 3.6 MeV/amu Au$^{53+}$ + He Collisions: A 4-Body Classical Treatment

**François Frémont** 

Université de Caen-Normandie, Centre Interdisciplinaire de Recherche sur les Ions, la Matière et la Photonique, 6 bd du Maréchal Juin, 14050 Caen Cedex, France; francois.fremont@ensicaen.fr; Tel.: +033-658057482

**Abstract:** A four-body classical model based on the resolution of Hamilton equations of motion was used here to determine and analyze ionization doubly-differential cross sections following 3.6 MeV/amu Au$^{53+}$ + He collisions. Our calculation was not able to reproduce the binary peaks experimentally observed in the transverse momentum distributions for electron emission energies larger than 10 eV. Surprisingly, by introducing a large number of free or quasi-free electrons that followed the projectile at the same velocity, the agreement between the experiment and our calculation was improved, since our model reproduced, at least qualitatively, the experimental binary peaks. The origin of the presence of such electrons is discussed.

**Keywords:** ion-atom collisions; single ionization; strong perturbation regime; Classical Trajectory Monte Carlo (CTMC) calculations; fully differential cross sections

## 1. Introduction

Since the early 1980's, much attention has been devoted to the scattering of fast ionized projectiles in ion-atom collisions. In particular, many experimental and theoretical studies have been performed on the particular cases of 100 MeV/amu C$^{6+}$ + He and 3.6 MeV/amu Au$^{53+}$ + He collisions [1–13], for which perturbation parameter $\eta = Z_p/v_p$ (i.e., the projectile charge-to-velocity ratio in atomic units are 0.1 and 4.4 a.u., respectively). The choice of these two collision systems has been motivated by the fact that, for highly charged ions at moderate velocities, the strong interaction between the outgoing projectile, the residual ionized He$^+$ target and the emitted target electron requires much more sophisticated theoretical treatments than that for C$^{6+}$ + He collisions.

Let us first focus on C$^{6+}$ + He collisions and, especially, doubly differential cross sections (DDCS) for the emission of one target electron (single ionization or SI) $d^2\sigma_{SI}/dq_\perp dE_e$, where $\vec{q}_\perp$ is the transverse momentum transfer of the projectile and $E_e$ is the emitted electron energy. Experimentally, DDCS strongly decrease when decreasing $q_\perp$ (i.e., when increasing the impact parameter b) [2]. At $E_e$ larger than 50 eV, corresponding to more violent encounters, a peak appears at transverse momentum transfer equal to the momentum of the emitted electron, showing that binary collisions between the electron and the projectile play a decisive role [2]. Experimental DDCS were found to be nicely reproduced by Born and Continuum Distorted Wave–Eikonal Initial State (CDW-EIS) calculations [2]. Classical Trajectory Monte Carlo (CTMC) 3-body calculations were also applied, neglecting electron-electron interaction (also called the electron correlation), and thus using effective charges for the He target [4]. Reasonable agreement with the experiment was observed, and the binary peak at large values of $q_\perp$ was also revealed. In addition, the absolute values of the calculated cross sections were in good agreement with the experiment.

For the Au$^{53+}$ + He system, for which similar structures corresponding to binary collisions appear [2], CDW and CTMC theoretical models [2,4], even after inclusion of nuclear-nuclear interaction,

were found to strongly deviate from the experiment. None of these methods were able either to reproduce the absolute magnitude of experimental cross sections or to show the presence of the binary peaks.

Afterward, different theories have been implemented to better describe experimental results. For example, 15 years ago, CDW-EIS calculations were presented for $Au^{53+}$ + He collisions [7]. The projectile-residual target $He^+$ interaction was considered by using the concept of effective Coulomb charge. Within the frame of this method, two different approaches were chosen. First, a fixed effective charge $Z_{eff}$ = 1.35 was used. Second, a scattering-angle-dependent effective charge describing the screening of the projectile by the passive electron was employed. Two-bend structures similar to the experimental data for the higher electron energies were observed. However, on the quantitative aspect, important differences were evidenced between experiment and theory. In addition, the model failed when comparing theory and experiment for triple differential cross sections (see for example Figure 3 of Ref [7]).

More recently, in [10], a detailed analysis of DDCS was improved using a CDW-EIS model. Different potentials were used to represent the interaction between the projectile, the emitted electron and the residual target $He^+$. However, whatever the chosen potential, the calculations could not reproduce the observed structure. The authors improved their model by including the so-called inelastic channel, that is, the dynamical excitation of $He^+$ during the collision. Thus, the passive electron was allowed to evolve. With this assumption, a better agreement with experiment was obtained but only for small values of the electron energy. As a conclusion, the authors suggested that only a four-body motion would allow to describe simultaneously all the DDCS features.

Unfortunately, a complete four-body description of a collision is difficult to implement using quantum mechanics, due to electron correlation. Nevertheless, since classical mechanics has been proven to be an alternative to quantum mechanics and give excellent qualitative and quantitative results in 4-body ion-atom collisions, at high or low impact energies [14,15], 4-body classical Monte Carlo (4B-CTMC) method was applied in this study to the collision system 3.6 MeV $Au^{53+}$ + He to determine $q_\perp$ distributions $d^2\sigma_{SI}/dq_\perp dE_e$. The advantage of the present method is to treat at the same time single ionization (SI), which is of interest here, and simultaneous target ionization and excitation (IE), according to previous calculations by Fainstein and Gulyás [10].

The method to calculate DDCS is briefly described in Section 2. The total ionization cross section was calculated and compared to available SI cross sections, and DDCS are then presented in Section 3. The presence of the binary peak is discussed in terms of collisions between electrons and the He target. It is shown that, despite improved agreement between present calculations and experimental results, the explanation of the presence of such electrons is a matter that is fraught with controversy.

## 2. Results

### 2.1. Classical Method

The CTMC method is based on a numerical solution of Hamilton's equations of motion for the many-body system, which includes in the present problem the $Au^{53+}$ and $He^{2+}$ nuclei, as well as both He electrons. The classical Hamiltonian reads:

$$H = \sum_{k=1}^{4} \frac{p_k^2}{m_k} + \sum_{k=1}^{3} \sum_{j=k+1}^{4} \frac{q_j q_k}{r_{jk}} + \sum_{i=1}^{2} \sum_{\beta=1}^{2} V_H^\beta(r_{\beta i}, p_{\beta i}) \tag{1}$$

In the above expression, $\vec{p_k}$, $q_k$ and $m_k$ are the momentum vector, the charge and the mass of particle $k$, respectively. The quantity $r_{jk}$ is the distance between particles $j$ and $k$. The pseudo-potential $V_H^\beta(r_{\beta i}, p_{\beta i})$, where $\beta$ denotes each nucleus and $i$ the index for each electron, was first introduced in nuclear physics [16] and then adapted for atom structures [17] and ion-atom [18] or ion-molecule

collisions [19]. The quantities $r_{\beta i}$ and $p_{\beta i}$ are the positions and momenta of He electrons, respectively, relative to the He nucleus or the Au nucleus. The expression of $V_H^\beta(r_{\beta i}, p_{\beta i})$ is

$$V_H^\beta(r_{\beta i}, p_{\beta i}) = \frac{\xi_H^2}{4\alpha\mu_\beta r_{\beta i}^2} exp\left[\alpha\left(1 - \left(\frac{r_{\beta i}\,p_{\beta i}}{\xi_H}\right)^4\right)\right] \qquad (2)$$

In this relation, $\mu_\beta$ is the reduced mass between each nucleus and one He electron. The quantities $\xi_H = 0.9582$ and $\alpha = 4$ were chosen so that the first and second ionization potentials of He are close to experimental ones. With these values, first and second ionization potentials (1.1 and 2.3 a.u., respectively) of He were consistent with the expected energies of 0.9 and 2 a.u.

Initially, the projectile was at a distance $z_p = -200$ a.u., and the orientation of the electron around the target was randomly chosen. The impact parameter $b$ varied from 0 to 20 a.u., and the angle $\varphi_p$, which characterized the position of the projectile in the $(xOy)$ plane, was also randomly chosen. The initial spatial and momentum distributions were calculated using the method initiated previously by Abrines and Percival [20] and developed in several cases for H target or multielectron targets (see for example [21]). More precisely, in the case of the He target, the electrons are in a symmetrical position compared to that of He nucleus, and their initial momentum vectors are opposite.

From the initial conditions, the Hamiltonian equations are numerically solved using the fourth-order Runge-Kutta method, with an adaptive step defined and described in Ref. [20]. At the end of the collision, the energy of the target electrons that had been ionized was determined as a function of the scattering angle $\theta_P$ and the impact parameter $b$. Thus, $d^2\sigma_{SI}/dq_\perp dE_e$ was deduced since $\theta_P$ is directly connected to $q_\perp$. To obtain good statistics, the number of calculated trajectories was fixed to 5,000,000.

### 2.2. $C^{6+}$ + He Collisions

Before analyzing $Au^{53+}$ + He collisions, it was necessary to verify that the present 4B-CTMC model was able to reproduce experimental DDCS in the case of 100 MeV/amu $C^{6+}$ + He collisions, for which previous theoretical models predict binary peaks, in accordance with experiment [2,8]. The Figure 1 shows DDCS for $C^{6+}$ + He collisions as a function of $q_\perp$, for emission electron energies $E_e = 10, 50, 90$ and 145 eV. The experimental cross sections (open circles) exhibit maxima for the three largest energies, due to binary collisions. For the record, first Born calculations (short dashed curves) [2] and CTMC calculations using independent He electrons (dashed curves) [8] are also plotted. Our calculated DDCS, convoluted with the experimental resolution of ~0.2 a.u., are in reasonable qualitative agreement with experiment. For $E_e = 10$ eV, our results are similar to the previous CTMC calculations, except at $q_\perp \sim$ 1.6 a.u., where a maximum appears, that cannot presently be explained. For $E_e = 50$ eV, agreement between 4B-CTMC calculations and experiment is good. Surprisingly, for larger values of $E_e$, despite the presence of binary peaks, our calculation noticeably deviates from the experiment and previous CTMC calculations, especially at small values of $q_\perp$. However, one has to note that, at these small values of $q_\perp$, our calculations are close to first Born calculations. At present, there is no explanation for this deviation. A structure, which is not explained to our knowledge, is observed experimentally at $q_\perp$ of the order of 1–1.2 a.u. Nevertheless, It is seen that, in our calculations, similar structures appear at $q_\perp$ of about 1.5 and 2 a.u. for electron energies of 50 eV and 145 eV, respectively. However, the overall observed concordance allows us to extend our calculations to $Au^{53+}$ + He collisions.

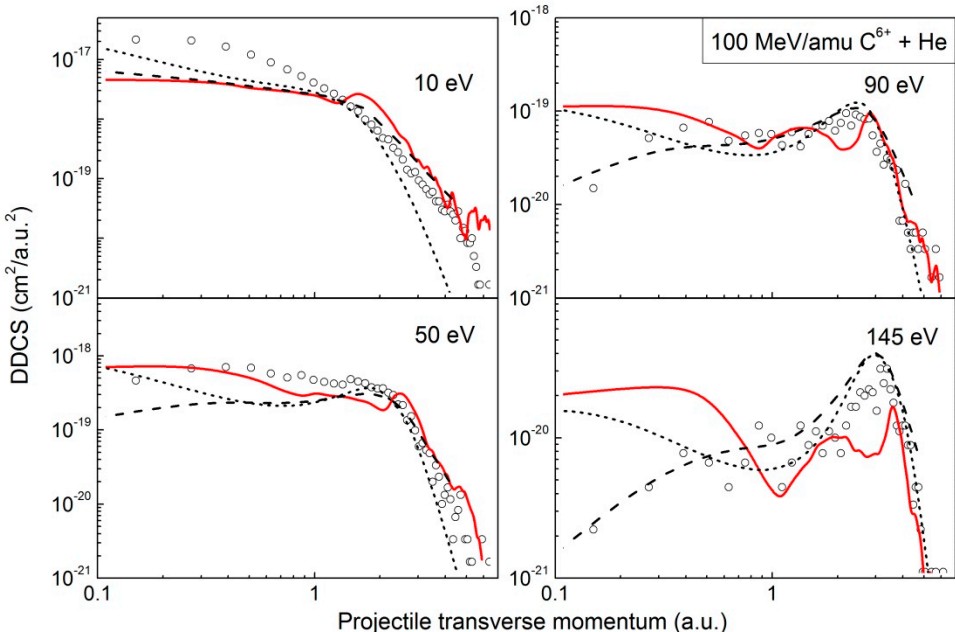

**Figure 1.** Doubly differential cross sections (DDCS) as a function of the projectile transverse momentum following 100 MeV/amu $C^{6+}$ + He collisions, for emission electron energies $E_e$ = 10, 50, 90 and 145 eV. Experiment, open circles [2]; CTMC calculations using independent electron model, dashed curves [2]; First Born calculations, short dashed curves; present four-body classical Monte Carlo (4B-CTMC) model, convoluted with the experimental resolution of ~0.2 a.u., full curve. The total cross section is normalized to $8 \times 10^{-15}$ cm$^2$.

### 2.3. $Au^{53+}$ + He Collisions

In Figure 2, the DDCS $d^2\sigma_{SI}/dq_\perp\, dE_e$ for single ionization of He by 3.6 MeV/amu $Au^{53+}$ projectiles are represented as a function of $q_\perp$ and for electron energies $E_e$ of 10, 50, 90 and 130 eV. The calculated total ionization cross section was found to be $7 \times 10^{-15}$ cm$^2$, which is close to the value of $8 \times 10^{-15}$ cm$^2$, according to previous experimental cross sections [22]. Once again, experimental cross sections (open circles) exhibited structures centered at $q_\perp$ larger than 1 a.u. Previous CTMC calculations (dashed curves) and CDW-EIS results (short dashed curves) were unable to reproduce the experimental DDCS. The present 4B-CTMC results (full curves) agree with the experiment for small values of $q_\perp$. However, the binary structures are not observed in our calculations.

As a first conclusion, the inclusion of electron correlation in classical calculations improves absolute DDCS, except at large values of $q_\perp$. Therefore, electron correlation is not the major contribution that would explain the differences between the experiment and the theories. At this stage, two hypotheses may be invoked: either the experiment is considered valid and something important is missing in all the theories or, on the contrary, the theories are more or less valid, depending on the approximations made, and the experiment is not valid or not complete. In the present discussion, the goal is not to decide the validity of experimental results or theoretical models. However, since binary peaks were only present in the experiment, the possibility that other particles may have interacted with the He target, giving rise to prominent maxima, is examined. Magnetic fields in the beam line separate charges, so that the probability that ion projectiles with charges different that 53 exist is negligible. Therefore, $Au^{q+}$ projectile with q < 53 are unlikely to participate to the collision. Thus one eventuality is that projectiles that may interact with the target are electrons. In the next section, collisions of electrons with He are studied, at projectile velocities equal to that of $Au^{53+}$. Then, the origin of such electrons will be discussed.

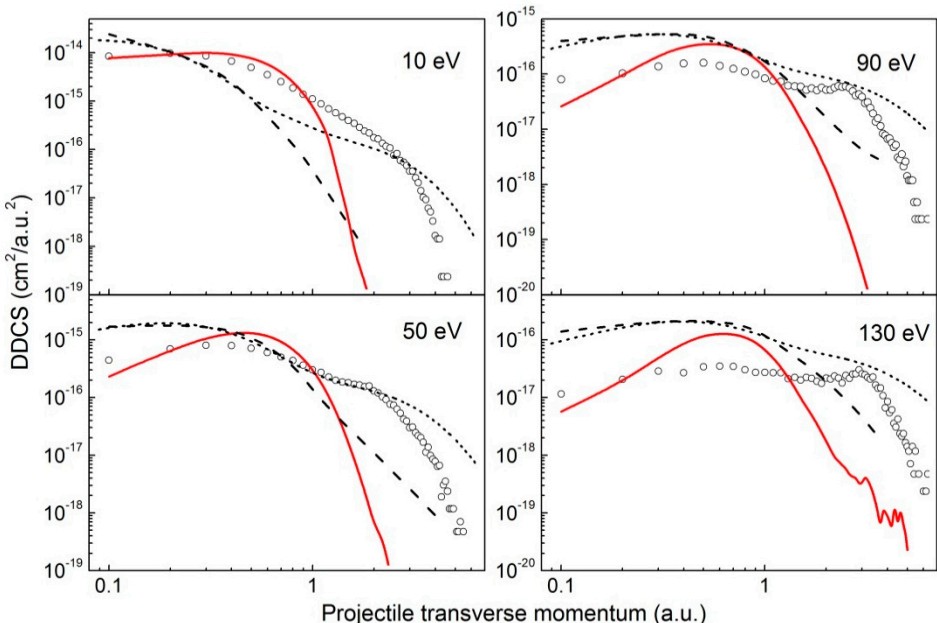

**Figure 2.** DDCS as a function of the projectile transverse momentum following 3.6 MeV/amu Au$^{53+}$ + He collisions, for emission electron energies $E_e$ = 10, 50, 90 and 130. Experiment, open circles [2]; CTMC calculations using independent electron model, dashed curves [8]; CDW-EIS calculations, short-dashed curves [2]; present 4B-CTMC model, convoluted with the experimental resolution of ~0.2 a.u., full curve.

## 2.4. Contribution of 1.96 keV e$^-$ + He Collisions

The single ionization DDCS following 1.96 keV e$^-$ + He collisions are presented in Figure 3 as a function of $q_\perp$ and for the same electron energies $E_e$ as previously mentioned. Note that the electron projectile energy has the same velocity as that of Au$^{53+}$, i.e., 12 a.u. Two different calculations were performed. First, all interactions were taken into account (dashed curves in Figure 3). Binary structures are observed but, surprisingly, their maxima are shifted to larger values of $q_\perp$ compared to maxima observed experimentally. In addition, a maximum for $E_e$ = 10 eV is revealed which was not present experimentally. Secondly, the interaction between the projectile and He nucleus was removed. The goal of this approximation is to reduce the scattering angle of the projectile. In this case, the position of the binary peaks maxima (full curves) is then in agreement with the experimental position of the maxima, and the binary peak at $E_e$ = 10 eV disappears.

However, DDCS in the case of electrons as projectiles are at least two orders of magnitude smaller than the experimental DDCS. Consequently, one has to invoke the contribution of more than 100 electrons as projectiles in order to quantitatively reproduce the binary contributions. The results are shown in Figure 4. The full curve is the sum of both Au$^{53+}$ (dashed curves) and e$^-$ (short-dashed curves) contributions, weighted by coefficients that are used to fit the experimental data. As mentioned above, the number of electrons necessary to reproduce the experiment was of the order of a few hundred.

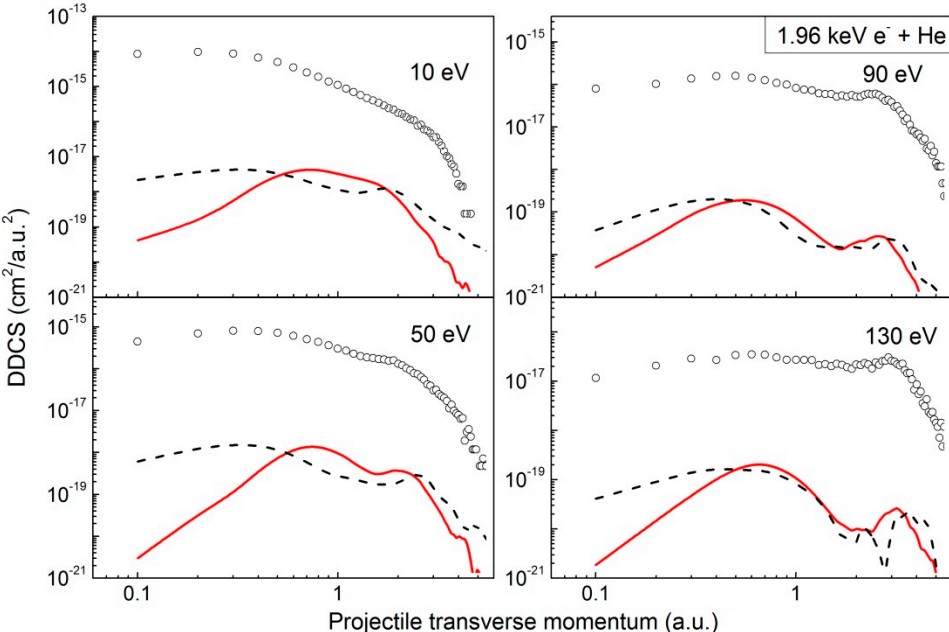

**Figure 3.** DDCS as a function of the projectile transverse momentum following 1.96 keV e⁻ + He collisions, for emission electron energies $E_e$ = 10, 50, 90 and 130 eV. Experiment for $Au^{53+}$ + He collisions, open circles [2]; present 4B-CTMC calculations, convoluted with the experimental resolution of ~0.2 a.u., dashed curve; present CTMC calculations neglecting interaction between e⁻ projectiles and He nucleus, full curve.

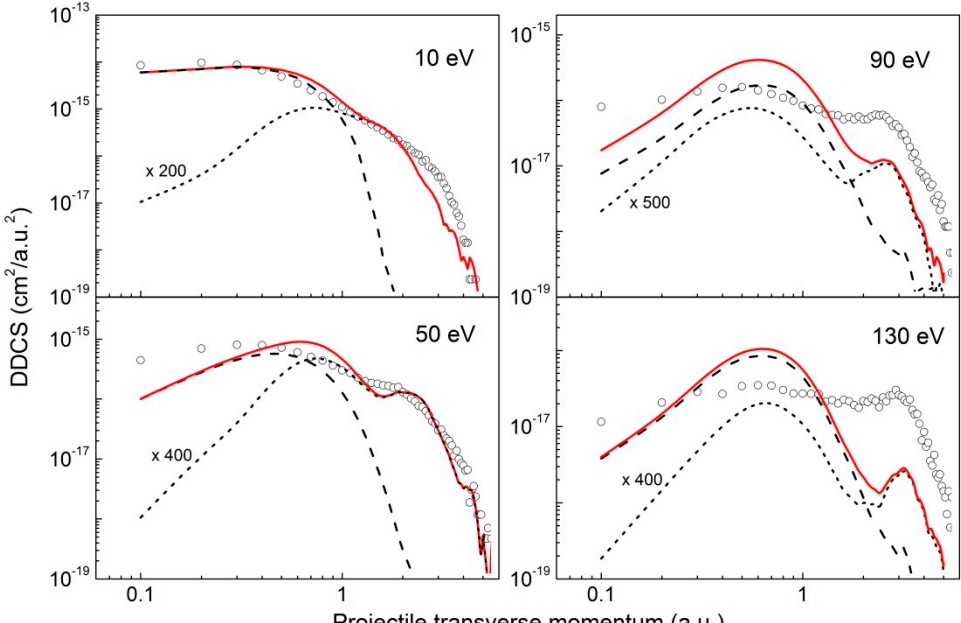

**Figure 4.** DDCS as a function of the projectile transverse momentum following 3.6 MeV/amu $Au^{53+}$ + He and 1.96 keV e⁻ + He collisions, for emission electron energies $E_e$ = 10, 50, 90 and 130. Experiment, open circles [2]; $Au^{53+}$ + He collisions, dashed curves; e⁻ + He collisions, short-dashed curves; some of both, full curves. The first two curves are weighted by coefficients in order to fit experimental data.

At 10 and 50 eV, agreement between calculations and experiment is rather good. The agreement is poorer at larger electron energies. Whatever the number of electrons involved, the calculated binary contribution is much smaller than the experimental one. The reason is probably that it is difficult to

correctly treat e$^-$ + He collisions using classical methods. At small values of $q_\perp$, the agreement is only qualitative, since the shape of experimental DDCS is not reproduced.

The origin of these electrons, if they really contributed to the present DDCS, is not evident. One would suggest that they come from collisions between metallic surfaces (slits for example) before penetrating the collision zone. It has been shown previously [23] that, for 5 MeV/amu ions penetrating carbon and insulator foils at normal incidence, the number N of emitted electrons, for large projectile charges q, is of the order of 0.3 q$^2$. For C$^{6+}$ ions, N ~ 10 while, for Au$^{53+}$ ions, N ~ 1000, which is the same order of magnitude as that found in the present calculation. However, according to the authors of [24], the convoy electrons are unlikely to collide with the He target, due to magnetic fields that strongly deviate electron trajectories. The present argument would be true if the electrons are really free. In fact, the electrons are not free, since they strongly interact with the Au$^{53+}$ projectile. Consider an electron of velocity equal to 13 a.u. in a magnetic field B =100 Gauss (top of Figure 5). The electron trajectory (dashed curve) is strongly deviated. Let us consider now an electron with a velocity close to the projectile velocity (12 a.u.), and in the same magnetic field. As seen in Figure 5 (bottom), when the electron (red curve) is at a distance of about 50 a.u. from the projectile (blue curve), the electron, despite the presence of the magnetic field, stays close to the projectile, with an helicoidal trajectory. Due to the presence of B, the radius increases, but very slowly. At a distance of 20 cm, the radius reaches a few thousands of a.u., which is much smaller than the width of the target jet, so that the electron is, in principle, able to reach the target.

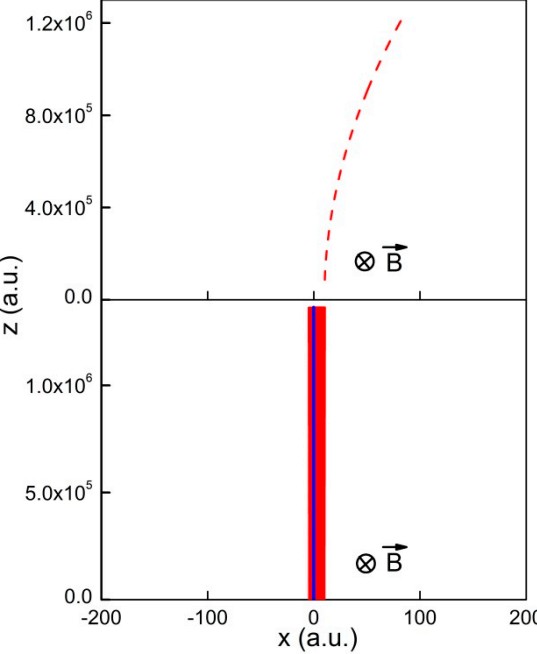

**Figure 5.** Top of the figure, electron trajectory (dashed red curve) in the presence of a magnetic field ; bottom of the figure, electron trajectory (full red curve) in the presence of a magnetic field and an electric field produced by an Au$^{53+}$ ion projectile. The electron is initially at a distance of about 50 a.u. of the projectile and its velocity is close to that of the projectile.

## 3. Conclusions

DDCS following single ionization of He by Au$^{53+}$ ions at a projectile energy of 3.6 MeV/amu were revisited using the CTMC method. Electron correlation was taken into account in the calculation. The agreement with experiment was found to be poor, since binary contributions observed experimentally were not reproduced by the present model. When including free electrons (i.e., independent of the projectile), the agreement with experiment was much better, especially at small emitted electron energies. One possibility is that these electrons are produced in collisions between Au$^{53+}$ ions and

surfaces before entering the collision zone. If the present hypothesis is valid, that is, if electrons are partly responsible for binary peaks, a strong dependence of DDCS with the charge of the projectile would be expected. Further experiments would be desirable to confirm or reject this hypothesis. On a theoretical point of view, CTMC calculations will be performed by considering electrons in the field of $Au^{53+}$ ions rather than free electrons.

**Funding:** This research received no external funding.

**Acknowledgments:** The author gratefully acknowledges M. Schulz, R. Moshammer and O. Fojon for fruitful discussions.

**Conflicts of Interest:** The authors declare no conflicts of interest

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
