# Peer review of "Transverse Momentum Transfer Distributions Following Single Ionization in 3.6 MeV/amu Au53+ + He Collisions: A 4-Body Classical Treatment"

_atoms, doi:10.3390/atoms6040068_

Round 1
Reviewer 1 Report
In this paper a previous study (2001, 2005) on transverse momentum transfer distributions following single ionization in two highly charged projectile - He collisions is revisited.
The aim of the paper was showing the improvement of performing a four-body classical calculation, to determine and analyze ionization doubly-differential cross sections in Au53+ + He collisions at 3.6 MeV/amu. The authors find that this approximation is useless and the agreement with the experiment is poor since the theoretical calculations still do not present the binary structure experimentally observed in the DDCS at electron emission energies greater than 10 eV. The authors then suggest the structure may be due to electrons emitted during the experiment that may interact with the target and therefore, CTMC calculations of collisions of electrons with helium atoms at 1.96 keV energy are performed. They find that the DDCS as functions of the projectile transverse momentum reproduce the experimental binary structure.
The idea is original, the study is correctly designed, the results seem to be appropriately interpreted and arouse interest for readers. Nevertheless, I have some general and technical comments that the authors should take into account.
In the Introduction, some incomplete sentences are found:
Line 61-62, I guess that line 62 has to continue the paragraph of line 61.
Line 67: ‘Unfortunately, … quantum mechanics’. The authors have to be more explicit and if necessary add something like ‘to describe ionization of highly charged projectile collisions’.
Lines 70: The authors should refer to intermediate energies instead of ‘at high or at low impact energies’, since it is well-known that the CTMC, 3 or 4 body treatments, works well in the intermediate impact energy range, especially regarding ionization processes.
In section 2.1, an illustration of the stabilization of the initial distributions for the He target atom in the phase space will be interesting to show or mention. The authors refers to Ref. [21], but this work is for one-electron CTMC calculations. It is well known that the He classical atom is unstable and it is necessary that the authors explain the method they have employed to stabilize the initial distribution in the 4-body calculation (see Jorge et al. Phys. Rev. A 94, 022710 (2016) and references therein).
In Figures 1 and 2, the authors present their results for the two studied collision systems compared to the experimental and other theoretical data. One can observe a very different behavior in both four-body CTMC calculations, a binary (multiple) peak structure appears in C6+ + He collisions while this peak structure is missed in Au53+ + He collisions. These differences should be discussed as well as the experimental explanation of the origin of the second peak at qperp ~ 1.2 – 1.3 au at emission electron energies > 50 eV, which is missing in all the text.
Respect to the two CTMC calculations, this work and that of Ref. [8], it is surprising the opposite behavior of the DDCS at small values of qperp, especially at at Ee > 90 and 150 eV, although the authors point out that the inclusion of electron correlation improves the absolute DDCS, some comments respect to this point are required.
The results shown in Fig. 3 need an extended explanation on how the e + He calculations have been performed, which independent electron model has been used and a how the cross section compares with experimental or theoretical ones at that high collision energies, if there are any.
The argument on line 195 remains unclear. A short discussion on the origin of those free electrons, which travel with the projectile (Fig. 5), will be welcome.
There are some misprints along the text:
Line 24, energies must be in MeV/amu instead of MeV.
Line 92, the second rβi must be pβi because it refers to the momenta of the He electrons.
Line 125, after calculations appears a strange word ‘Guillaumeo’ that has to be removed.
Line 146, check the spelling of the word ‘correlation’.
Line 182. Caption of Fig. 4, ‘sum of both’ instead of 'some'.
Author Response
response to first referee
Comments and Suggestions for Authors
In the Introduction, some incomplete sentences are found:
Point 1 : Line 61-62, I guess that line 62 has to continue the paragraph of line 61.
In fact, I don’t really understand what the referee wants to say.
Point 2 : Line 67: ‘Unfortunately, … quantum mechanics’. The authors have to be more explicit and if necessary add something like ‘to describe ionization of highly charged projectile collisions’.
I just added “due to electron correlation”. Whatever the process (electron capture or ionization, a four-body problem is so difficult using quantum mechanics that, in most of the cases, independent models are used.
Point 3 : Lines 70: The authors should refer to intermediate energies instead of ‘at high or at low impact energies’, since it is well-known that the CTMC, 3 or 4 body treatments, works well in the intermediate impact energy range, especially regarding ionization processes.
I replaced “high or low energies” by intermediate or low energies”, since recent publications show that, at low energies, 4B CTMC calculations give rise to good results.
Point 4 : In section 2.1, an illustration of the stabilization of the initial distributions for the He target atom in the phase space will be interesting to show or mention. The authors refers to Ref. [21], but this work is for one-electron CTMC calculations. It is well known that the He classical atom is unstable and it is necessary that the authors explain the method they have employed to stabilize the initial distribution in the 4-body calculation (see Jorge et al. Phys. Rev. A 94, 022710 (2016) and references therein).
I didn’t want to show a figure in order not to complicate the subject. The method has been described in this publication. The stability is given by the pseudopotential V and the initial conditions are written in the text : “More precisely, in the case of He target, the electrons are in a symmetrical position compared to that of He nucleus, and their initial momentum vectors are opposite.”
In Figures 1 and 2, the authors present their results for the two studied collision systems compared to the experimental and other theoretical data. One can observe a very different behavior in both four-body CTMC calculations, a binary (multiple) peak structure appears in C6+ + He collisions while this peak structure is missed in Au53+ + He collisions. These differences should be discussed as well as the experimental explanation of the origin of the second peak at qperp ~ 1.2 – 1.3 au at emission electron energies > 50 eV, which is missing in all the text.
First, I changed a sentence, in order to say that the agreement is qualitative in general. “Our calculated DDCS, convoluted with the experimental resolution of ~0.2 a.u., are found to be in qualitative agreement with experiment.” The origin of the structure at qperp ~ 1.2 – 1.3 au is not explained in the experimental publication. I added a sentence : “A structure, which is not explained to our knowledge, is observed experimentally at of the order of 1 – 1.2 a.u. Nevertheless, it is seen that, in our calculations, similar structures appear at of about 1.5 and 2 a.u. for electron energies of 50 eV and 145 eV, respectively.”
Respect to the two CTMC calculations, this work and that of Ref. [8], it is surprising the opposite behavior of the DDCS at small values of qperp, especially at at Ee > 90 and 150 eV, although the authors point out that the inclusion of electron correlation improves the absolute DDCS, some comments respect to this point are required.
Yes, it is surprising. This is the reason than I wrote, in the original version : “For larger values of Ee, despite the presence of binary peaks, our calculation noticeably deviates from experiment and previous CTMC calculations, especially at small values of .” I just inserted “At present, there is no explanation for this deviation.”
The results shown in Fig. 3 need an extended explanation on how the e + He calculations have been performed, which independent electron model has been used and a how the cross section compares with experimental or theoretical ones at that high collision energies, if there are any.
The calculations were performed as in the previous sections. First, a 4B CTMC calculation was used, secondly, the projectile He nucleus interaction was removed. So I changed a little bit the sentence : “Secondly, the interaction between the projectile and He nucleus was removed. The goal of this approximation is to reduce the scattering angle of the projectile.” Unfortunately, to my knowledge, there is no experimental or theoretical DDCS for this system to compare.
The argument on line 195 remains unclear. A short discussion on the origin of those free electrons, which travel with the projectile (Fig. 5), will be welcome.
I don’t understand this remark. To my opinion, the tentative of explanation is clear : “One would suggest that they come from collisions between metallic surfaces (slits for example) before penetrating the collision zone.” Note that I added “(slits for example) in the text.
There are some misprints along the text:
Line 24, energies must be in MeV/amu instead of MeV. corrected
Line 92, the second rβi must be pβi because it refers to the momenta of the He electrons. corrected
Line 125, after calculations appears a strange word ‘Guillaumeo’ that has to be removed. corrected
Line 146, check the spelling of the word ‘correlation’. corrected
Line 182. Caption of Fig. 4, ‘sum of both’ instead of 'some'. corrected
Reviewer 2 Report
This manuscript presents another attempt to theoretically explain long existing data for single ionization of helium by fast, highly charge gold ions.
Here, a 4-body classical treatment is used. The author is well recognized as an experienced theoretician using this and other methods. The present method is compared to experimental data and to other existing theoretical treatments for both
the Au53+ - He of interest and for a test system of C6+ - He.
Although the present 4-body treatment is argued to work well for the test system, I find
some disturbing deviations and the agreement with experiment to be poorer than existing theories. (See additional comments below.) The 4-body treatment is
then applied to the Au53+ - He system of interest where, compared to the existing theories shown, it is only better for very small transverse projectile
momentum transfers. For large values of qperp it provides a much poorer comparison with experiment. However, the main point of this manuscript is that
this 4-body classical treatment also yields no evidence of the binary structure seen in the experimental data. The author attempts to explain this by
proposing that a large number (several hundred) free electrons accompany the incoming gold ion and simultaneously interact with the target. Doing so yields
a peak in the binary location but overall this is only qualitative agreement with experiment and to achieve quantitative agreement appears very unlikely.
In addition, that several hundred free electrons are somehow produced and interact with the target simultaneously with the highly charged ion is, in my
opinion, extremely unlikely and would need to be supported by information from the original authors of the experimental paper.
Specific comments include:
lines 119-120: From viewing the figure, "good agreement" seems overly generous. A better description for the comparison of the present CTMC calculation and experiment is: "are found to
be in reasonable qualitative agreement with experiment."
line 125: There is some typo problem/error on this line. Please correct.
Regarding Fig. 1: It is disturbing that the present theory yields a definite binary structure at 10 eV where
experiment shows no evidence of such structure and that at higher electron energies it yields a double peaked binary structure which
noone expects. Considering that the major point of this paper is with regard to the observed binary peak in
Au53+ -He collisions and that the C6+ - He system was used as a confirmation of the theory, the statements about
confidence in extending the calculations to Au53+ are troubling.
Fig. 2: again the vast discrepancy between the present CTMC calculations with previous theories, one of which is a CTMC calculation, is troubling.
The discussion following this figure is reasonable with respect to why no theory is able to reproduce a binary peak,
but it mostly neglects why the present theory is so different/poor for larger qperp.
Lines 154-55: The statement that only electorns may interact may make sense to the author, but to this reviewer, does not.
Magnetic fields will strongly influence electrons. Please explain how this statement is deduced.
If figures 3 and 4 are to support the idea that "free" electrons account for the observed binary structure, I am not
convinced. Even after including an extremely large number of electrons there is virtually no agreement above 50 eV, e.g.,the binary peak is still
an order of magnitude too small while at the same time the calculated intensity in the region of 0.5-1 a.u. is too large by a factor of 5 or so. Plus
the number of free electrons is not internally consistent.
Lines 184-200: Regarding the discussion about the number of electrons produced from impact on surfaces, the vast majority of these electrons have energies less than 20 eV. There are virtually no electrons in the the hundred eV, let alone the keV range.
Electrons having trajectories and energies as described could be produced by stripping (ionizing) the projectile, in this case it would most like be Au52+, i.e.
single ionization of the projectile. These would be produced at low energy in the projectile frame and would therefore be the required velocity in the lab frame.
They would also primarily be in the direction of the projectile.
That said, how and why would so many electrons be produced? The cross section would be extremely small compared to the cross sections shown for ionizing the target. This would imply an extremely large contaminant of the beam and require a stripping medium,
that would not influence the 53+ component. These problems are not adressed here but need to be in order to accept the arguements and conclusions presented. As stated above, such arguements would need to be supported by statements from the authors
of the experimental work.
Author Response
Second referee
This manuscript presents another attempt to theoretically explain long existing data for single ionization of helium by fast, highly charge gold ions.
Here, a 4-body classical treatment is used. The author is well recognized as an experienced theoretician using this and other methods. The present method is compared to experimental data and to other existing theoretical treatments for both the Au53+ - He of interest and for a test system of C6+ - He.
Point 1 : Although the present 4-body treatment is argued to work well for the test system, I find some disturbing deviations and the agreement with experiment to be poorer than existing theories. (See additional comments below.) The 4-body treatment is then applied to the Au53+ - He system of interest where, compared to the existing theories shown, it is only better for very small transverse projectile momentum transfers. For large values of qperp it provides a much poorer comparison with experiment. However, the main point of this manuscript is that this 4-body classical treatment also yields no evidence of the binary structure seen in the experimental data. The author attempts to explain this by proposing that a large number (several hundred) free electrons accompany the incoming gold ion and simultaneously interact with the target. Doing so yields a peak in the binary location but overall this is only qualitative agreement with experiment and to achieve quantitative agreement appears very unlikely.
The referee is right. For C6+ + He, the agreement is only qualitative. However, the binary peaks are observed. And again, he is right concerning Au53+ + He collisions. At present, there is no quantitative agreement between experiment and our calculations. We have to keep in mind that those are preliminary calculations, assuming free electrons. In a further paper, we will show that improvement may be observed when assuming electrons in the field of Au53+ projectile.
Point 2 : In addition, that several hundred free electrons are somehow produced and interact with the target simultaneously with the highly charged ion is, in my opinion, extremely unlikely and would need to be supported by information from the original authors of the experimental paper.
I discussed with some experimentalists who claim that it is not possible that electrons may disturb the collision. However, I prove at the end of the manuscript that the simultaneous presence of a magnetic field and Au53+ only give rise to a very small perturbation of the electron trajectory (see lines 197 to 218 of the manuscript), since the convoy electrons are close to the projectile.
Specific comments include:
Point 3 : lines 119-120: From viewing the figure, "good agreement" seems overly generous. A better description for the comparison of the present CTMC calculation and experiment is: "are found to be in reasonable qualitative agreement with experiment."
Yes, I do agree with the referee. Consequently I changed the sentence.
Point 4 : line 125: There is some typo problem/error on this line. Please correct. corrected
Regarding Fig. 1: It is disturbing that the present theory yields a definite binary structure at 10 eV where experiment shows no evidence of such structure and that at higher electron energies it yields a double peaked binary structure which no one expects. Considering that the major point of this paper is with regard to the observed binary peak in Au53+ -He collisions and that the C6+ - He system was used as a confirmation of the theory, the statements about confidence in extending the calculations to Au53+ are troubling.
The referee is right. I have no explanation why this structure at 10 eV appears. For this reason, I added (lines 121 – 122) : "where a maximum appears, that cannot be explained at present"
Point 5 : Fig. 2: again the vast discrepancy between the present CTMC calculations with previous theories, one of which is a CTMC calculation, is troubling.
The referee is partly right. The discrepancy is obvious at large Ee, but not at 10 eV or 50eV. Secondly, at small qper values and large Ee, our calculations noticeably deviate from experiment. This is also seen in the case of first Born calculations. For these reasons, I changed the sentences : "For Ee = 50 eV, agreement between 4B-CTMC calculations and experiment is good. Surprisingly, for larger values of Ee, despite the presence of binary peaks, our calculation noticeably deviates from experiment and previous CTMC calculations, especially at small values of . However, one has to note that, at these small values of , our calculations are close to first Born calculations. At present, there is no explanation for this deviation. A structure, which is not explained to our knowledge, is observed experimentally at of the order of 1 – 1.2 a.u. Nevertheless, I t is seen that, in our calculations, similar structures appear at of about 1.5 and 2 a.u. for electron energies of 50 eV and 145 eV, respectively. However, the overall observed concordance allows us to extend our calculations to Au53+ + He collisions."
Point 6 : The discussion following this figure is reasonable with respect to why no theory is able to reproduce a binary peak, but it mostly neglects why the present theory is so different/poor for larger qperp.
The referee is right. But this is the goal of the paper to show that other processes may occur, if electron correlation cannot explain the presence of binary peaks. I added the following sentence : " Therefore, electron correlation is not the major contribution that would explain the differences between experiment and the theories. " (lines 152-153)
Lines 154-55: The statement that only electrons may interact may make sense to the author, but to this reviewer, does not. Magnetic fields will strongly influence electrons. Please explain how this statement is deduced.
It was also a surprise that electron could be present during the collision. But, as I said in point 1, the Au53+ field is much stronger that the magnetic field, so that electrons can be present during the collision for a long time. The referee is right, concerning the lines 154-155. To soften, I changed the sentence : “Thus, it is possible that the only projectiles that may interact with the target are electrons.”
If figures 3 and 4 are to support the idea that "free" electrons account for the observed binary structure, I am not convinced. Even after including an extremely large number of electrons there is virtually no agreement above 50 eV, e.g.,the binary peak is still an order of magnitude too small while at the same time the calculated intensity in the region of 0.5-1 a.u. is too large by a factor of 5 or so. Plus the number of free electrons is not internally consistent.
As I pointed out in the text, the production of such a number of electrons having the same velocity as that of the projectile has been previously studied by many authors. According to Schiewitz et al, the number of electrons of the same velocity as that of Au53+ may reach ~1000. So, what is important here is the order of magnitude of the number of electrons necessary to explain the presence of binary peaks. In addition, the referee is right. A classical description of e- + He collisions is not obvious, so that we don't know how correct is our DDCS. As pointed out before, there are no DDCS experimental or theoretical results to compare with.
Lines 184-200: Regarding the discussion about the number of electrons produced from impact on surfaces, the vast majority of these electrons have energies less than 20 eV. There are virtually no electrons in the the hundred eV, let alone the keV range.
It is well known that most of the electrons ejected by surfaces have a low energy. However, it has been clearly shown by different authors (for example Schiewitz, cited in the manuscript) that the number of electrons of the same velocity as the projectile is large in the case of highly charged ions impacting on surfaces.
Electrons having trajectories and energies as described could be produced by stripping (ionizing) the projectile, in this case it would most like be Au52+, i.e. single ionization of the projectile. These would be produced at low energy in the projectile frame and would therefore be the required velocity in the lab frame.
They would also primarily be in the direction of the projectile.
That said, how and why would so many electrons be produced? The cross section would be extremely small compared to the cross sections shown for ionizing the target. This would imply an extremely large contaminant of the beam and require a stripping medium, that would not influence the 53+ component. These problems are not adressed here but need to be in order to accept the arguments and conclusions presented. As stated above, such arguments would need to be supported by statements from the authors of the experimental work.
Stripping the projectile is very hard, since the electrons have high binding energy. I can't imagine that these electrons would be provided. Finally, I cannot agree with the referee. Our arguments are only hypothesis, that have to be confirmed or not. To my knowledge, there is no proof in papers describing the present experiment that electrons are absent during the collision. This has to enter into an internal debate, to my opinion. To conclude, since, again, I just propose an hypothesis, which is coherent, since convoy electrons were proven (Figure 5) to keep their trajectory, despite of the magnetic field, more experiments would be desirable to confirm or not the presence of such electrons. Of course, refined theoretical calculation are also desirable.

Reviewer 3 Report
Report of the Referee -- "Transverse momentum transfer distributions following single ionization in 3.6 MeV/amu Au53+ + He
collisions : A 4-body classical treatment", by F. Frémont
----------------------------------------------------------------------
The article presents double diferential cross sections (ddcs) calculations for Au53+ +He and C+6+He collisions, for similar values of the
perturbation parameter Zp/vp. The main attempt is to shine some light over the failure of quantum (born and cdw-eis) and classical
(ctmc) theories to describe Au53+ +He results, while they are able to describe C6+ +He cross sections. A four-body ctmc calculation
including the fnal bounded He electron is presented, which is able to mainly describe structures of the C6+ +He ddcs in a better way
than the quantum and classical three-body approximations. After the failure of all the theories to describe the Au53+ +He collisions at
large electron emission energies, the contribution o 1.96 keV convoy electrons following the projectile is considered, which seem to
produce the adequate structures but with lower magnitude. The hypothesis is that a quantity of a few hundred electrons are needed to
reproduce experimental results.
Some of the hypothesis and approximations presented in the paper would be a matter of a deeper discussion. However, they are presented
with the adequate precaution and clarity, according to the complexity of the problem.
I recommend the article for publication after the author answer some questions and make minor corrections.
Questions and corrections:
* In the frst paragraph of the introduction, from line 26 to 29: "The choice of these two collision..." From that sentence it is clear why the
Au53+ collision was chosen, but not why the C6+ was. Maybe the role of the similar value of the perturbation parameter, both in the
physics of the problem and in the perturbative approaches has to be commented.
* Line 41 to 24: "even at large Ee". I do not understand why the agreement should be worse at large Ee than for small Ee, where
quantum behavior becomes important.
* Line 92: should be a "p" letter for the momenta.
* Line 125: "tGuillaumeo".
* Line 135 and 136: "cross section" -> "cross sections".
Author Response
Third referee
The article presents double diferential cross sections (ddcs) calculations for Au53+ +He and C+6+He collisions, for similar values of the perturbation parameter Zp/vp. The main attempt is to shine some light over the failure of quantum (born and cdw-eis) and classical
(ctmc) theories to describe Au53+ +He results, while they are able to describe C6+ +He cross sections. A four-body ctmc calculation including the final bounded He electron is presented, which is able to mainly describe structures of the C6+ +He ddcs in a better way than the quantum and classical three-body approximations. After the failure of all the theories to describe the Au53+ +He collisions at large electron emission energies, the contribution o 1.96 keV convoy electrons following the projectile is considered, which seem to produce the adequate structures but with lower magnitude. The hypothesis is that a quantity of a few hundred electrons are needed to reproduce experimental results. Some of the hypothesis and approximations presented in the paper would be a matter of a deeper discussion. However, they are presented with the adequate precaution and clarity, according to the complexity of the problem. I recommend the article for publication after the author answer some questions and make minor corrections.
Questions and corrections:
* In the first paragraph of the introduction, from line 26 to 29: "The choice of these two collision..." From that sentence it is clear why the Au53+ collision was chosen, but not why the C6+ was. Maybe the role of the similar value of the perturbation parameter, both in the
physics of the problem and in the perturbative approaches has to be commented.
In fact, to be clearer, I changed "has been motivated" by 'was motivated". The referee is true, I only chose Au53+ + He, while the experimentalists chose both C6+ and Au53+ + He collisions.
* Line 41 to 24: "even at large Ee". I do not understand why the agreement should be worse at large Ee than for small Ee, where quantum behavior becomes important. The referee is right, so I removed these words
* Line 92: should be a "p" letter for the momenta. corrected
* Line 125: "tGuillaumeo". corrected
* Line 135 and 136: "cross section" -> "cross sections". corrected

Round 2
Reviewer 2 Report
The primary message of this manuscript is that by including a large (but from figure 4, an arbitrarily variable) number of
fast free electrons, long standing discrepancies between experimental data and theoretical calculations can be understood.
But, in my opinion, there is no evidence to support the origin of these electrons. Reference 23, which the author uses to
support this idea is a study of transmission through thin foils where the total electron yield is used to show the large
number of electrons used here to provide qualitative agreement with experiment. But the total electron emission
includes both foreward and backward emission. From one study of the measured foreward/backward ratios, e.g. Rothhard et al, PRA 41, (1990),
a reasonable estimate for the presently discussed energy and ion is this ratio is 3-5. Although this would still allow the large number of electrons
used in Fig. 4, the vast majority of these electrons have low energies; only a very small portion have the desired energy. Also,
the studies in ref 23 were performed using transmission through thin foils, not grazing incidence collisions which would happen by the
ions at the edge of the beam passing through a collimating slit/aperture. Third, as noted by the author in his response, his discussions with experimentalists have not supported
this suggestion. Therefore, although desired to perhaps explain existing discrepancies between experiment and theory, the source
of these electrons needs to be supported. This, in my opinion, has not been done. That is why in my original review, I suggested that the original experimentalists
need to provide supporting input for the present idea to be valid.
A second difficulty in the paper, which still is unresolved, is the (in many cases) poor agreement between the present theory with the data
and previous theories. The author has weakened various statements about the "agreement" plus inserted various qualifying
statements, but the lack of agreement still exisits. In the author's response to this, he noted that these are preliminary calculations
and that a further paper is planned. If so, what purpose does the present paper have? Could submission not be delayed until
"final" calculations are available? [My apologies to the author for these blunt statements but in light of the fact that a
followup paper is planned and that the topic of discussion is not new which makes it unlikely that time is of essence, I really
think the problems mentioned above need to be answered prior to publication.]
Therefore, my recommendation is for rejection at present and resubmission after a) final calculations are available and b) strong supporting
evidence for the source of the large number of electrons needed is available.
Author Response
Second referee
The primary message of this manuscript is that by including a large (but from figure 4, an arbitrarily variable) number of fast free electrons, long standing discrepancies between experimental data and theoretical calculations can be understood. But, in my opinion, there is no evidence to support the origin of these electrons. Reference 23, which the author uses to support this idea is a study of transmission through thin foils where the total electron yield is used to show the large number of electrons used here to provide qualitative agreement with experiment. But the total electron emission includes both forward and backward emission. From one study of the measured forward/backward ratios, e.g. Rothard et al, PRA 41, (1990), a reasonable estimate for the presently discussed energy and ion is this ratio is 3-5.
If we carefully read the article by Rothard et al., it has to be mentioned first that the foils are penetrated by singly charged ions (Ne+, … Xe+), and not highly charged ions. The difference of charges would, may be, change the number of secondary electrons. Second, figure 2 in this mentioned article shows that, for Xe+ projectile, the number of forward electrons is about 20 at the lowest projectile energy and 33 at the highest energy (30keV/u), which is much smaller than the present energy (3.6MeV/amu). We can expect that the number of secondary electrons would be much larger at the latter energy.
Although this would still allow the large number of electrons used in Fig. 4, the vast majority of these electrons have low energies; only a very small portion has the desired energy.
According to the publication I cited (Schiewitz), it is not true.
Also, the studies in ref 23 were performed using transmission through thin foils, not grazing incidence collisions which would happen by the ions at the edge of the beam passing through a collimating slit/aperture.
At present, it is hard to say if the Au53+ projectile go through the foil or collide the slit involving grazing incidence.
Third, as noted by the author in his response, his discussions with experimentalists have not supported this suggestion. Therefore, although desired to perhaps explain existing discrepancies between experiment and theory, the source of these electrons needs to be supported. This, in my opinion, has not been done. That is why in my original review, I suggested that the original experimentalists need to provide supporting input for the present idea to be valid.
I cannot agree with this. When I spoke with experimentalists, they didn't say that there were not secondary electrons. They only said that, if secondary electrons exist, they are deviated because of the magnetic field. However, I have clearly shown that, with a magnetic field of 100 Gauss, which is much larger than the magnetic field of ~20 Gauss the experimentalists use, in principle, the secondary electrons are not deviated, because of the presence of the strong presence of Au53+.
A second difficulty in the paper, which still is unresolved, is the (in many cases) poor agreement between the present theory with the data and previous theories.
Yes, the agreement is poor at present. Nevertheless, the referee has to admit that the agreement is not bad at electron energies of 10 eV and 50 eV. For the higher energies, I just said that the agreement is qualitative.
The author has weakened various statements about the "agreement" plus inserted various qualifying statements, but the lack of agreement still exists. In the author's response to this, he noted that these are preliminary calculations and that a further paper is planned. If so, what purpose does the present paper have? Could submission not be delayed until "final" calculations are available? [My apologies to the author for these blunt statements but in light of the fact that a follow up paper is planned and that the topic of discussion is not new which makes it unlikely that time is of essence, I really think the problems mentioned above need to be answered prior to publication.]
In the present paper, the conclusion is clear. I say "On a theoretical point of view, CTMC calculations will be performed by considering electrons in the field of Au53+ ions rather than free electrons." The referee has to understand that, first, calculations including Au53+ projectile, He2+ nucleus, two correlated electrons, and, let say, 100 secondary electrons, require a huge amount of computational time ! And the analysis of the data has to be carefully done, in order to be sure that there is no mistake in the calculations themselves. In addition, parameters have to be changed in order to see a possible dependence of the results with these parameters. The present work is a preliminary one. I am aware of how this work has to be improved. But the solution I give is original and has to be taken into account, to my opinion.
Therefore, my recommendation is for rejection at present and resubmission after a) final calculations are available and b) strong supporting evidence for the source of the large number of electrons needed is available.